# Above- and Below-Ground Carbon Sequestration in Shelterbelt Trees in Canada: A Review

**Rafaella C. Mayrinck \***[ID]**, Colin P. Laroque, Beyhan Y. Amichev**[ID] **and Ken Van Rees**

School of Environment and Sustainability, University of Saskatchewan, Maintenance Road, 110, Saskatoon, SK S7N 5C5, Canada; colin.laroque@usask.ca (C.P.L.); beyhan.amichev@vt.edu (B.Y.A.); kcv903@mail.usask.ca (K.V.R.)

\* Correspondence: rcm786@mail.usask.ca; Tel.: +1-306-850-2717 or +55-15-9-9798-5054

**Abstract:** Shelterbelts have been planted around the world for many reasons. Recently, due to increasing awareness of climate change risks, shelterbelt agroforestry systems have received special attention because of the environmental services they provide, including their greenhouse gas (GHG) mitigation potential. This paper aims to discuss shelterbelt history in Canada, and the environmental benefits they provide, focusing on carbon sequestration potential, above- and below-ground. Shelterbelt establishment in Canada dates back to more than a century ago, when their main use was protecting the soil, farm infrastructure and livestock from the elements. As minimal-and no-till systems have become more prevalent among agricultural producers, soil has been less exposed and less vulnerable to wind erosion, so the practice of planting and maintaining shelterbelts has declined in recent decades. In addition, as farm equipment has grown in size to meet the demands of larger landowners, shelterbelts are being removed to increase efficiency and machine maneuverability in the field. This trend of shelterbelt removal prevents shelterbelt's climate change mitigation potential to be fully achieved. For example, in the last century, shelterbelts have sequestered 4.85 Tg C in Saskatchewan. To increase our understanding of carbon sequestration by shelterbelts, in 2013, the Government of Canada launched the Agricultural Greenhouse Gases Program (AGGP). In five years, 27 million dollars were spent supporting technologies and practices to mitigate GHG release on agricultural land, including understanding shelterbelt carbon sequestration and to encourage planting on farms. All these topics are further explained in this paper as an attempt to inform and promote shelterbelts as a climate change mitigation tool on agricultural lands.

**Keywords:** agroforestry systems; carbon sequestration; climate change mitigation; windbreaks; shelterbelts

---

## 1. Overview: Shelterbelt Qualities and Their Role around the World

Shelterbelts, also known as windbreaks, are agroforestry systems that can be defined as barriers of trees, or trees combined with shrubs, that are planted to reduce wind speed [1–4]. Hedges are a similar feature, defined as a narrow row of a low and dense shrub species used to separate fields [4]. Sometimes shelterbelt and hedge concepts can be interchangeable, because shelterbelts are also used to separate fields and hedges end up reducing wind speed. However, in this paper, we will be focusing on the aspects of shelterbelts only.

Shelterbelts have been established all over the world to protect soil, crops, homes, farm infrastructure, livestock, and pastures. In Britain, shelterbelts were largely planted in the mid-18th century for crop protection and to keep farm pollution away from busy roads [5]. In the United States (U.S.), a shelterbelt-incentive program was carried out by the Prairie States Forestry Project (PSFP), which resulted in nearly 30,000 km of shelterbelts planted from 1935 to 1942 across six Great

Plains states [6]. In China, shelterbelts have been used to isolate the coastal zone from sea and land disturbances [7], to protect agricultural systems from dry winds and sandstorms [8], and to stabilize sand dunes [9]. In 1950, an extensive shelterbelt planting took place, aimed at defeating agricultural lands from erosion [7]. Later, the "Three-North Shelterbelt Project" started, and has increased treed land area from 5%, in 1978, to 10%, in 2008 [9]. In New Zealand, landowners have planted shelterbelts since 1850, when the settlers arrived, totaling more than 300,000 km in length [10]. In Australia, shelterbelts were planted on treeless areas such as the western plains of Victoria [11]. In Argentina, there are more than 1500 km of windbreaks planted to protect crops, cattle and homes from wind [12].

Shelterbelts can be composed of perennial and or annual trees and shrubs [1–3]. The species chosen should be adapted to local climate, topography, and soil [13]. To make it sustainable through time, it is recommended to alternate rows of fast and slow-growing species [14], creating a forest-like dynamic. Fast-growing species start protecting the area earlier allowing the slow-growing species to reach maturity when the fast-growing species are in decline, thus always maintaining an effective shelter. This system enriches biodiversity, while producing wood that can be harvested periodically for fencing, furniture and housing, as well as increasing carbon residence time in the system. Combining species within the overall design makes the shelterbelt system sustainable through time, as well as making the system more resistant to pests and disease, diversifying shelterbelt structure and assisting to mitigate any of its vulnerabilities [3,13].

Ideal shelterbelt structure and design depend on its function [2]. For example, for wind protection, it should have multiple rows (usually 2–3) of trees to achieve high shelterbelt density, located at 2–5 times the shelterbelt height (H) from the edge of the field, to increase the amount of land protected and amplify economic returns [1,2,13]. For snow management, normally, the ideal design is planting one single row with a tall deciduous tree species using wide spacing (5 to 7 m between trees to achieve a medium shelterbelt density), perpendicular to the prevailing winds [2]. In many cases, there are other directions that winds can put crops in danger, rather than the prevailing winds, so it is beneficial to have two right-angle oriented shelterbelt rows [13]. For severe winters, as in the Canadian Prairies, five to seven rows may provide the ideal protection from weather events [13]. For livestock systems, shelterbelt rows should be dense, planted at narrow spacing (2–3 m between trees), so that the animals are protected from associated wind chill [2]. Normally, one windbreak is not enough to protect a whole field, so more rows need to be added, within a distance of 10–20 times the shelterbelt height, depending on the level of protection desired, size of equipment used, and degree of crop tolerance to wind [2,15]. For example, Helmers and Brandle [16] recommended to add a shelterbelt every 13 H for corn and soybean production in a 70-year planning horizon.

An important factor on establishing shelterbelts is row spacing indicated by the distance between planted trees within a shelterbelt row. If narrow spacing is adopted, the trees will shade the soil beneath much sooner, which can reduce the costs of weed control; however, the disadvantage is that trees will be competing earlier for resources, and if not managed properly, could lead to reducing their health and growth [13]. Wider spacing also has disadvantages, since the trees will take longer to shade the soil to avoid weed competition, so the landowner will have to combat weeds; the trees will develop larger crowns, demanding more water; greater tree and soil exposure to the sun and wind between the rows will also increase evapotranspiration. Between shrubs, conifers and deciduous trees the minimal recommended spacing is 0.3–1, 2.0–2.5 and 3 m, respectively [17].

Shelterbelt maintenance is similar to forest maintenance. At their establishment, weeding and pest control should be conducted, to help seedlings get established. If the shelterbelt is planted in a pasture or an area with wild animals, building a fence should be considered to protect the seedlings from animal injuries [14,17]. Later, branch pruning and thinning of densely planted rows may be required in some cases, to boost height and diameter growth, respectively [14].

The efficiency of a shelterbelt as a wind barrier is determined by its external and internal structure. Its external structure is related to its height, length, orientation, continuity, width, and cross-sectional shape; its internal structure is related to the amount and structure of open and solid spaces in the tree

crowns, plant shape, and surface area [1–3]. Shelterbelt height and length determine the extent of windbreak protection [1,3]. Shelterbelt length should be 10 [1–3] to 20 [15] times its height, to reduce wind flow around the ends of the shelterbelt [1–3].

As shelterbelt use and importance become more popular around the world, more research on the topic has been published. Figure 1A illustrates the percentage of existing literature published per year, available on the Web of Science, on the topic of shelterbelts and the respective percentages of published papers per year on the topics of agriculture and forestry. It was expected that agriculture and forestry publication rates would surpass shelterbelt publication rates for three main reasons. First, agriculture and forestry are crucial to feed the increasing demand of the growing global population. Second, shelterbelts are normally not established with the intention of producing material goods as they are for agriculture and forestry. Third, the environmental benefits provided by shelterbelts are only mostly experienced in the long run and are difficult to be translated into monetary values, unlike the products from agriculture and forestry. However, shelterbelt publications were increasing at a pace similar to agriculture and forestry rates, rising sharply after the 90s (Figure 1A). This may be attributed to the increasing access to computers, making it easy to process studies and publish more, or perhaps to the increasing awareness on climate change and environmental issues. In either case, awareness on the importance of shelterbelts and their environmental services was increasing, regardless of their smaller role on providing material goods. Similar searches were made using the keyword windbreak, instead of shelterbelt which yielded the same trend as shown in Figure 1A. The total contribution to shelterbelt research by country varied (Figure 1B). The top three leading countries on shelterbelt research are (in decreasing order) China, U.S., and Canada.

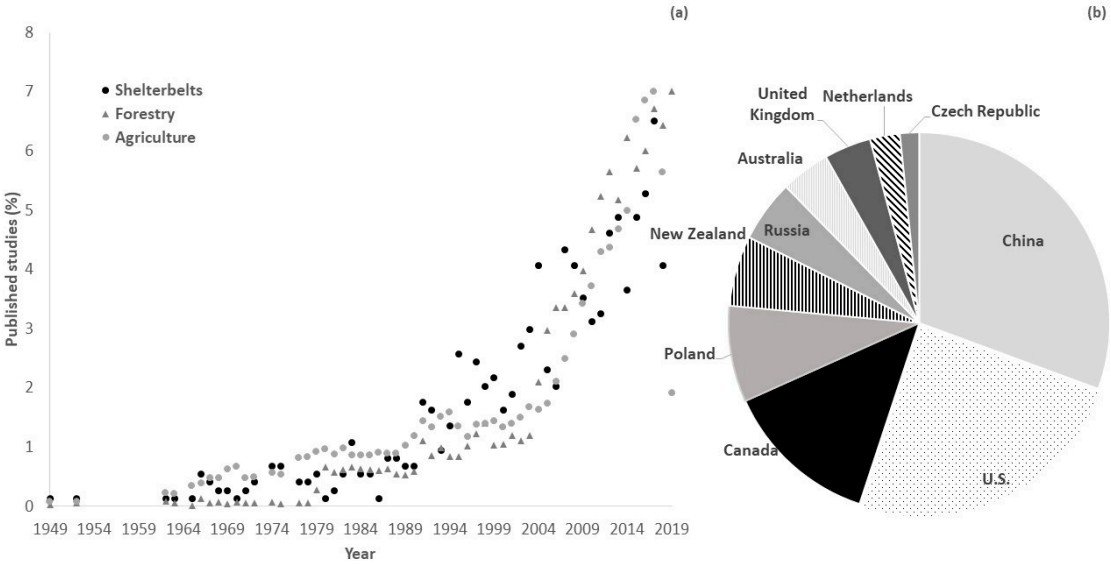

**Figure 1.** Number of journal publications on shelterbelt agroforestry systems found on Web of Science (**a**), and the relative contribution by country (**b**).

Shelterbelt researchers in Canada have contributed much to the literature due to the great number of shelterbelts planted over the years. According to the Government of Canada, farmers in Western Canada have planted more than 600 million trees during the past century [18]. Just in Saskatchewan alone, the total length of shelterbelts equals to 60,633 km [19], sequestering around 4.85 Tg C in the past nine decades, more than half of which were sequestered since 1990 (3.77 Tg C) [20].

The motivations driving farmers around the world to plant shelterbelts have been in general agreement: to protect farm yards, crops, infrastructure, and livestock from the harsh environment, and for aesthetics [1,2,5]. However, this general motivation has changed over the years to address the needs of each generation of farmers; thus, new reasons for planting and maintaining shelterbelts were added. For example, in New Zealand, shelterbelts planting was also focused on timber production [21].

In Canada, as described above, shelterbelts have been planted mainly to protect soil from wind erosion; however, because of new agricultural techniques, such as the no-till system, wind erosion is less of an issue than in past decades, and some farmers are currently removing their shelterbelts [1,2]. In a recent study, it was determined that 29.8% of farmers in a 1400 km$^2$ study area in Saskatchewan, Canada, removed their shelterbelts between 2008 and 2016 [21]. The main reasons that farmers were removing their existing shelterbelts were: (1) shelterbelts required additional labor for maintenance; (2) shelterbelts made it difficult to operate large farming equipment in crop fields; and (3) shelterbelts reduced the land area available for crop production [21–24].

However, the benefits provided by the shelterbelt may be able to overcome even these new disadvantages. Some researchers have suggested that farmers are not aware of the advantages of having shelterbelts, so, they are more prone to remove them [23–25]. It is very likely that if education on the environmental benefits that shelterbelts provide, mainly their carbon sequestration, were projected into the long term, it would help farmers to better value them. Rewarding farmers for keeping existing, and or, planting new shelterbelts, by for example granting them a tax reduction or tax credit in a carbon trading market place [21], would make landowners more prone to stop removing them. If a carbon market were implemented, any carbon sequestered by shelterbelt trees would have a monetary value, so trees would become a part of the farm's overall budget, which could motivate future shelterbelt planting and better maintenance of existing ones.

## 2. Shelterbelt History in the Canadian Prairies

When European settlers first started to populate Western Canada in the late 1890s, they were encouraged to settle in various regions, including the driest area, known as the Palliser Triangle [24,26]. This area encompasses southern Saskatchewan, into Alberta, and Manitoba including the Canada–U.S. border, covering more than 200,000 square kilometers [24,26]. There is even a dryer area inside the Palliser Triangle, named the Dry Belt, located in southern Alberta and Saskatchewan [24,26]. This land is extremely arid, and has had many cycles of extreme dry years through time [26].

Between the 1880s and 1980s, several droughts occurred in the Canadian Prairies: in 1910, 1914, 1917–20, 1924, 1929–30 [6,26]. During these dry periods, shelterbelts were useful in capturing snow, and helped increase and maintain soil moisture. Planted trees improved the soil water regime by shading and reducing soil surface temperatures [6], all of which reduce soil evapotranspiration. At the time, there was also a false belief that planting trees would increase local rainfall [6,26], which contributed to increases in local shelterbelt planting. The 1930s drought occurred during one of the worst crises in North America, known as the Great Depression. It was a combination of drought, insect infestation, and dropping of global commodities prices [26]. The Great Depression was so brutal that, in 1936, about 14,000 people left their farms, totaling around 12,140 km$^2$ of abandoned land within the Palliser Triangle alone [6,26].

To aid farmers in Western Canada facing Depression-like conditions, the Government of Canada established the Prairie Farm Rehabilitation Administration (PFRA) in 1935, in Indian Head, Saskatchewan, where 200 hundred agrologists, engineers, field husbandmen, inspectors and administrators focused on rehabilitating farms in the Prairies after the Great Depression [26]. The PFRA received $750,000 for the first year, and $1 million annually for the next four years, to help combat the consequences of drought [26]. By 1939, the PFRA assisted with the construction of thousands of dugouts and earthen dams designed for stocking water [26]. Additionally, the PFRA assisted farms in many ways, including the management of community pastures in Saskatchewan and Manitoba, and by conducting a soil survey covering 90% of the Palliser Triangle, at a one-mile resolution [26].

One of the most notable legacies of the PFRA activity included providing free seedlings and assistance for shelterbelt establishment to farmers, known as the Prairie Shelterbelt Program (PSP) [27,28]. A nursery in Indian Head was created to function as a live demonstration farm to show planting options, and serve as a reference for farmers displaying the diversity of shelterbelt agroforestry systems that could be implemented on the Prairies [6]. Initially, seedling demand was low,

but this increased through time, from 1000 to 9.2 million trees per year, peaking in the particular years of 1961, 1970, 1981 and 1991, probably due to landowners receiving shelterbelt-related information from PFRA (i.e., hand-outs, talks at agricultural fairs and shows) related to droughts that occurred in the previous years [28]. As the demand for seedlings changed, the number of species offered also changed, increasing from five, at the beginning of the program, to 37 shelterbelt species in the 2000s [28]. It is calculated that in total, over 600 million shelterbelt trees/shrubs were provided by the PSP program [28,29]. Towards the end of the program, the demand for shelterbelt trees dropped, and in 2006 and 2009, 3.7 million and 2.5 million seedlings were ordered, respectively [28]. In the end, the PSP program was shut down in 2013.

The work promoted by PFRA's PSP program on implementing shelterbelts for the past nine decades showed Canada's commitment to sustainable development in the Prairies. This commitment was reaffirmed in 2009 by signing the Copenhagen Accord, when Canada committed to reduce GHG emissions by 17% by 2020, based on 2005 levels. To help with reaching the goal, the Federal government launched the Agricultural Greenhouse Gases Program (AGGP) in 2013. The goal of AGGP is to mitigate GHG emissions in the agricultural sector by creating technologies and practices that promote carbonless agriculture [30]. One of the aims of AGGP is to study shelterbelt carbon sequestration and their potential for climate change mitigation. This branch financed the majority of studies on shelterbelts in Canada since its inception. In 2016, the Paris Accord was signed by 170 countries, including Canada, which required committed countries to join efforts to keep global temperature increases under 2 °C, based on pre-industrial levels, with further efforts aimed at limiting warming to 1.5 °C [31]. As stipulated by the AGGP program, and in the context of global efforts towards a carbonless economy, the carbon sequestration potential of shelterbelts remains a viable research priority for the Canadian Prairies.

## 3. Environmental Services Provided by Shelterbelts

Shelterbelts provide both public and private benefits, commonly referred to as environmental services [32]. Private benefits include protecting soil, homes, farm infrastructure, and livestock from the elements [23,28,32–34], reducing animal odor from livestock systems, lowering the risk of crop environmental damage due to pesticide spray-drift [24,28,35], reducing noise [13], and heating costs for households and livestock operations [7,28]. It is estimated that shelterbelts can save up to 18% in energy costs for heating homes [36].

Public benefits provided by shelterbelts include reducing soil runoff into rivers, streams and creeks, sequestering carbon dioxide from the atmosphere, enhancing and protecting animal and plant diversity, including pollinators, as well as improving water quality [18,32,37]. Some shelterbelt benefits can be categorized as both private and public. For example, improving water quality and protecting biodiversity are more commonly classified as public environmental services; however, these can also be considered as private benefits since it would also be beneficial on the land where the shelterbelt is located.

These benefits are hard to quantify in terms of monetary value [18,24,32], due to the complexity of the factors affecting, and being affected by, shelterbelt systems, but there are some studies that address this question. Kulshreshtha et al. [32] assessed public service worth provided by shelterbelt seedlings given by PFRA's PSP program from 1981 to 2001. They found that those benefits were worth $140 million, the majority provided by carbon sequestration ($73 million), and soil erosion reduction ($15 million). Similarly, Amichev et al. [20] studied carbon sequestration of six common shelterbelt species in Saskatchewan, planted from 1925 to 2009, and estimated that the carbon additions (through $CO_2$ sequestration) in shelterbelt systems since 1990 (equal to 3.77 Tg C) would be worth $208 million dollars at current carbon prices.

Because shelterbelts can improve their surrounding conditions through environmental services, they can also impact crop production by retaining soil moisture, slowing wind speed, shading areas beside the trees, and reducing soil loss [1,15,38]. Shelterbelts can increase monetary gains for

landowners by increasing crop yield and/or saving on chemical applications [20,24,32]. Normally, these benefits can span up to a distance of 10 H on the leeward and 0–3 H on the windward side of the trees [38]. For example, it was calculated that shelterbelts planted from 1981 to 2001 in the Prairie Provinces in Canada, prevented soil erosion, equal to a benefit of $15 million [32]. Shelterbelts were also calculated to reduce crop production costs related to the use of pesticides, and reduce overall crop loss due to pest damage [39,40]. Gámez-Viruez et al. [40], studying feces from birds inhabiting shelterbelts in Australia, found that birds fed on crop pests, helping as a biological control. Shelterbelts can also help to keep a more stable soil temperature range in its surroundings. During the night, soil temperatures near shelterbelts are from 1–2 °C higher than in open fields, which can assist crops to germinate and grow faster in colder environments [1].

Baldwin [38] conducted a review on the effect of shelterbelts on crop production and concluded that gains can be up to 50%. Kort [41] completed an extensive review on how the crops respond to shelterbelt trees and found that in most cases, shelterbelts increased crop yield, and that this can be further maximized by choosing adequate shelterbelt species and designs, based on specific crops. The author concluded that wheat (*Triticum aestivum* L.), barley (*Hordeum vulgare* L.), rye (*Secale cereal* L.), millet (*Pennisetum americanum* L.), alfalfa (*Medicago sativa* L.), and hay (mixed grass and legumes) yields are more responsive to shelterbelt presence, and that oats (*Avena sativa* L.), and maize (*Zea may* L.) are affected as well, though they are less responsive. Hawke and Tombleson [10] observed a 15% increase in pasture production on both sides of shelterbelts at distances equal to 70% of the tree height.

The general rule is that shelterbelts have positive impacts on adjacent crop yield [1,15,38,42], which are more pronounced during short-duration or more intense droughts [25,41,43,44]. However, there are cases where shelterbelts impact yield negatively. Land trade-off is one factor leading to decreasing yield since shelterbelts take out land area from crop production [41]. To make shelterbelts economically viable, crop yield increases by a shelterbelt's presence should be more than compensated for the yield lost in the area that is used to plant the shelterbelt. In the long run, the benefits from the shelterbelt will therefore be felt economically [44]. According to Brandle et al. [2] shelterbelts are economically viable if less than 6% of the land is occupied by trees.

Another factor that may decrease crop yield is allelopathy on the adjacent crop, as it can inhibit seed germination and overall crop growth. Another factor is shading, created by the trees, that can reduce crop photosynthesis and growth from reduced sunlight [6,15,18,37]. For example, Kowalchuk and Jong [45,46] assessed the effect of shelterbelts on wheat yield and soil erosion for three years and found that when environmental conditions were dry, trees and crops competed for moisture, and yield was reduced at distances up to 10 m from the shelterbelt edge. Singh and Kohli [47] studied the effect of an eight-year-old *Eucalyptus tereticornis* Sm. shelterbelt on yields of chickpea (*Cicer arietinum* L.), lentil (*Lens culinaris* (LENCU) wheat, cauliflower (*Brassica oleracea* L.) and barseem (*Trifolium alexandrinum* L.) and concluded that yield was always reduced by shelterbelts and that chickpeas were the most affected crop from the list.

Table 1 shows the effect of shelterbelts on crops reported in the literature. It varies from case to case and can be positive or negative for the same crop, depending on many environmental factors, varying from year-to-year environmental inputs. In most cases, shelterbelt impact was positive. In the cases of decreasing yields, the results were mainly attributed to below-ground tree/crop competition for soil moisture and nutrients. One alternative to reducing below-ground competition is pruning the lateral tree roots [2,48,49]. The frequency of root pruning was dependent on the shelterbelt and crop species and spacing, but is normally done every one to five years [2,42]. It was calculated that in North America, root pruning can decrease competition from 10 to 44% in the adjacent shelterbelt-influenced area [2,38]. Onyewotu et al. [49] found that millet yield increased after pruning roots at 0.25 H distance from *Eucalyptus camaldulensis* Dehnh shelterbelt roots beside the field. Lyles et al. [42] also found that yield from 1 to 2 H distance were 1.6 higher than on the unpruned zone. However, root pruning is an expensive operation. To avoid root pruning, it is important to choose species with deep root systems, so they do not spread laterally and compete with crop root systems [2,42,43]. Greb and Black [43]

found that American elm (*Ulmus Americana* L.), black walnut (*Juglans nigra* L.), ponderosa pine (*Pinus ponderosa* Douglas ex P. Lawson & C. Lawson, and Siberian elm (*Ulmus pumila* L.) have shallow roots, so they spread laterally in search of nutrients, and compete more with crops. Thevs [50] found that among the shelterbelt species tested (tamarack (*Tamarix*), Siberian elm, Russian-olive (*Elaeagnus angustifolia* L.), honeysuckle (*Lonicera)*, and caragana (*Caragana arborescens* Lam., tamarack had the highest soil water uptake and caragana had the lowest, and therefore was a better suited species to be used in a shelterbelt composition.

**Table 1.** Effect of shelterbelts on adjacent crop yield from studies around the world.

| Shelterbelt Species | Crop Species | Influence | Yield (%) | Reference | Location |
|---|---|---|---|---|---|
|  | Winter wheat | + | 23 | Kort [a] [41] |  |
|  | Spring wheat | + | 8 | Kort [41] |  |
|  | Barley | + | 25 | Kort [41] |  |
|  | Oat | + | 6 | Kort [41] |  |
|  | Rye | + | 19 | Kort [41] |  |
|  | Millet | + | 44 | Kort [41] |  |
|  | Alfalfa | + | 99 | Kort [41] |  |
| Ponderosa pine | Winter wheat | + | 4.19 | Greb and Black [43] | Colorado, U.S. |
| Caragana, Chokeckerry (*Prunus virginiana* L.) | Winter wheat | + | 12.25 | Greb and Black [43] | Colorado, U.S. |
| Black walnut, Black locust (*Robinia pseudoacacia* L.) | Winter wheat | – | 12.89 | Greb and Black [43] | Colorado, U.S. |
| Ponderosa pine | Sorghum | – | 2.4 | Greb and Black [43] | Colorado, U.S. |
| Siberian pea, Chokeckerry | Sorghum | – | 13.5 | Greb and Black [43] | Colorado, U.S. |
| Black walnut, Black locust | Sorghum | – | 3.75 | Greb and Black [43] | Colorado, U.S. |
| (Populus × euramericana) | Soybeans (*Glycine max* L.) | + | 23 | Qi et al. [25] | Great Plains, U.S. |
| Green ash, Austrian pine (*Pinus nigra*), Eastern red cedar (*Juniperus virginiana* L.) | Soybeans | + | 26 | Ogbuehi and Brandle [51] | Nebraske, U.S. |
| Eastern red cedar | Beans | + | 21 | Rosenber [52] | Nebraske, U.S. |
| Indian rosewood (*Dalbergia sissoo*) | Cotton (*Gossypium hirsutum* L.) | + | 10 | Puri et al. [53] | Dhiranvas, India |
| *Corymbia intermedia* (R.T.Baker) K.D.Hill & L.A.S.Johnson, *Corymbia tessellaris* (F.Muell.) K.D.Hill & L.A.S.Johnson | Potato (*Solanum tuberosum* L.) | + | 6.7 | Sun and Dickinson [54] | Atherton Tablelands, Australia |
| Arizona cypress (*Cupressus arizonica*) |  | – | 25 | Campi et al. [55] | Rutigliano, Italy |
| Aleppo pine (*Pinus halepensis* Miller) | Wheat | + | 44 | Nuberg et al. [56] | Southern Australia |
| Aleppo pine | Faba beans (*Vicia faba* L.) | + | 49 | Nuberg et al. [56] | Southern Australia |
| Aleppo pine | Oat | + | 25 | Nuberg et al. [56] | Southern Australia |
| *Populus canadensis* Mönch, *P. Beijingensis* W.Y. Hsu, *P. xiaozuanrica, P. Simonii* Carrière, *P. pseudo-simonii* | Mayze | + | 6.13 | Zheng et al. [57] | Western China, Heilongjiang, Jilin, Liaoning and Inner Mongolia |

[a] Shelterbelt species and location are not provided because this numbers were summarized by Kort [41] after an extensive literature review from variety of studies (including a variety of locations and shelterbelt species).

Even though shelterbelt tree roots can compete with crops, they impact the overall ecosystem in a positive manner, due to their significant role on soil health and structure. It is calculated that global land degradation annually costs about $300 billion U.S. dollars [58]. A study conducted in England and Wales illustrated that soil compaction alone was responsible for 39% of all costs of soil recovery [59]. Soil compaction is one of the most serious issues faced by agricultural producers

today [34]. Soil compaction is caused by a variety of factors, such as overuse of heavy machinery and short crop rotations, which reduces crop yield and soil health in the long term. In contrast, shelterbelt systems can ameliorate crop growing conditions by improving soil structure and adding organic matter into the soil by means of growing extensive and deep tree root systems, thus enhancing soil porosity, which increases soil water infiltration and recharge, and improving the overall soil health. Carrol et al. [34] studied the effect of shelterbelts in pastures and observed that water infiltration under and near shelterbelts was 60 times greater compared to open areas on the pasture, and that significant changes in the rate of soil water infiltration happened soon after planting, as early as two years after shelterbelt establishment.

Besides water infiltration, crop water-use efficiency is also affected by shelterbelts. The treed barrier reduces wind speed, thus slowing heat transfers from the crops to the air, and slowing down evapotranspiration. Davis and Norman [60] reviewed the effects of shelterbelts on crop water-use and found a significant reduction in turbulent air which decreased evapotranspiration, improving water-use efficiency. Similarly, Ogbuehi [51] found that sheltered (shaded) soybeans had greater photosynthesis rate, stomatal conductance and deeper light penetration than non-sheltered plants. Thevs et al. [50], studying corn, potato, wheat, and barley production, found that crop water consumption was 10–12% lower in areas in the proximity of shelterbelts, compared to open field conditions.

Environmental services provided by shelterbelts, as previously discussed, modify the micro environment, and can be seen as a useful tool to mitigate the effects of climate change that will impact agricultural production worldwide in the near future [1,44]. For example, Easterling et al. [44] used a model to simulate the effects of climate change related stress on maize planted in dry environments in Nebraska, comparing yield on sheltered and unsheltered crops. They concluded that sheltered crops yields were greater, and that shelterbelts are an important tool to ameliorate global warming consequences.

## 3.1. Shelterbelt Carbon Sequestration Potential

Shelterbelts are useful not just to mitigate local weather extremes caused by climate change, but are also useful tools to mitigate global warming, thanks to their carbon sequestration potential. The Inter-Governmental Panel on Climate Change indicated that there were about 630 million hectares of unproductive land on the planet in 2000, and they suggested that if it was used for agroforestry, it would sequester 1.43 and 2.15 Tg $CO_2$ every year by 2010 and 2040, respectively [61].

Carbon sequestration in agroforestry varies among the different types of agroforestry practiced, ecological regions where it takes place, and soil type, ranging from 0.29 to 15.21 Mg ha$^{-1}$ year$^{-1}$ for above-ground, and 30–300 Mg C ha$^{-1}$ year$^{-1}$ up to 1 m depth in the soil [62]. Currently, the global area under agroforestry systems is 1023 million ha [63] and as previously stated, there is approximately 630 million ha of unproductive lands in the world that could be used to promote carbon sequestration through agroforestry practices [61]. Carbon sequestration potential tends to be greatest in natural forests, then in agroforestry systems, followed by tree plantations, and finally in cropped lands [49,63]. Table 2 shows the land use and its carbon sequestration potential reported in the literature around the world. Land use systems that include trees have a great potential in comparison with agriculture alone. Schoeneberger [64] and Wang and Feng [65] values are already in hectares, and they considered that shelterbelts occupied 5 and 2.5% of the cropland area, respectively.

Table 2. Carbon sequestration potential of diverse land use classes using different tree species around the world.

| Land Use | Species | Total Mg C ($km^{-1}$ $year^{-1}$) | Above-ground Mg C ($ha^{-1}$ $year^{-1}$) | Below-ground Mg C ($ha^{-1}$ $year^{-1}$) | Total Mg C ($ha^{-1}$ $year^{-1}$) | Location | Reference |
|---|---|---|---|---|---|---|---|
| Shelterbelt | Hybrid Poplar | 6.03–6.54 | 0.79*Total | 0.21*Total | 3.3–5.2 | Saskatchewan, Canada | Amichev et al. [a] [20] |
| Shelterbelt | Scots Pine (*Pinus sylvestris* L.) | 1.90–2.17 | 0.90*Total | 0.10*Total | 1.4–3.3 | Saskatchewan, Canada | Amichev et al. [20] |
| Shelterbelt | Manitoba Maple | 2.39–2.60 | 0.80*Total | 0.20*Total | 2.8–5.3 | Saskatchewan, Canada | Amichev et al. [20] |
| Shelterbelt | White Spruce (*Picea glauca* Moench) | 2.43–2.75 | 0.81*Total | 0.19*Total | 2.2–4.1 | Saskatchewan, Canada | Amichev et al. [20] |
| Shelterbelt | Green Ash | 1.78–1.98 | 0.77*Total | 0.23*Total | 2.0–3.9 | Saskatchewan, Canada | Amichev et al. [20] |
| Shelterbelt | Caragana | 1.73–2.03 | 0.74*Total | 0.26*Total | 1.3–2.7 | Saskatchewan, Canada | Amichev et al. [20] |
| Shelterbelt | | | 0.58-1.17 | | | Nebraska | Schoeneberger [b] [64] |
| Shelterbelt | *Poplar canadensis* Mönch, *Paulownia elongala* | | 0.38 | | | China | Wang and Feng [c] [65] |
| Woodlots for firewood, fodder, land reclamation | | | 1.0–5.0 | 1.0–6.0 | 2.0–11 | Asia/Africa | Nair et al. [63] |
| Shade tree system | *Cordia alliodora* (Ruiz & Pav.) Oken, *Theobroma cacao* L. | | 3 | | | Costa Rica | Montagnini et al. [66] |
| Shade tree system | *Erythrina poeppigiana* (Walp.) O.F. Cook*, Theobroma cacao* L. | | 4.4 | | | Costa Rica | Montagnini et al. [66] |
| Forest | *Dipteryx panamensis* (Pitt.) Rec. & Mell | | 20.26 | | | Costa Rica | Montagnini et al. [66] |
| Monoculture | Cooffe (*Coffea arabica* L.) | | 2.14 | | 2.14 | Brazil | Palm et al. [67] |
| Imporved fallow | *Crotalaria grahamiana* Wight & Arn | | 8.5 | 2.7 | 11.2 | Kenya | Albrecht et al. [68] |
| Imporved fallow | *Eucalyptus saligna* Sm. | | 21.7 | 9.55 | 31.25 | Kenya | Albrecht et al. [68] |
| Silvipasture | | | 6.1 | | 6.1 | North America | Udawatta and Jose [69] |
| Alley crop | | | 3.4 | | 3.4 | North America | Udawatta and Jose [69] |
| agriculture | Fallow, Soybean, Maize | | 0.3 | | 0.3 | Goias, Brazil | Bayer et al. [70] |
| Intercroping | Gliricidia sepium ((Jacq.) Steud) | | | 12.3 | 12.3 | Zomba, Malawi | Makumba et al. [71] |

[a] Reported per-area data (Mg $ha^{-1}$ $year^{-1}$) in Amichev et al. [20] represent the area directly underneath the live shelterbelt tree crowns. For comparison purposes in this study, we assumed that the area of the shelterbelts in Amichev et al. [20] represented 5% of the total farm area, similar to Schoeneberger [64]. [b] It was estimated that the area of shelterbelts represented 5% of the total farm area. [c] It was estimated that the area of shelterbelts represented 2.5% of the total farm area. #*Total: value times the total biomass (Total Mg C ($ha^{-1}$ $year^{-1}$)).

3.1.1. Shelterbelt Carbon Sequestration Potential and Stocks Above-Ground

Carbon sequestration has been extensively discussed as one of the main strategies to keep atmospheric carbon dioxide at acceptable levels, and minimize environmental risks from climate change effects. Given the increased awareness about shelterbelt carbon sequestration, many studies have been conducted in this regard, across a variety of climatic and edaphic regions, and across different shelterbelt designs, and species mixes [9,10,20,26,29,30,33,34,39,45,72]. The means to quantify carbon sequestration in shelterbelt systems have improved over the years, starting with the use of simple linear relationships and yield tables [73], to more sophisticated, and more accurate methods, using complex modelling frameworks, such as Holos, CBM-CFS3 (Carbon Budget Model for the Canadian Forest Sector) [37,74–76], and 3PG (Physiological Principles Predicting Growth) models [74].

The earliest research on shelterbelt carbon stocks in Canada was carried out by Kort and Turnock [73]. They used destructive sampling techniques to measure shelterbelt tree biomass, and fitted linear models to predict above-ground biomass for different shelterbelt species, ranging from 17 to 90 years, across all soil zones in the Saskatchewan Prairies. They reported an average biomass of 79 kg tree$^{-1}$ (32 Mg km$^{-1}$) for green ash, 263 kg tree$^{-1}$ (105 Mg km$^{-1}$) for hybrid poplar, and 144 kg tree$^{-1}$ (41 Mg km$^{-1}$) for white spruce. Some early work was also done to understand the interaction between shelterbelts and the adjacent crops in terms of C sequestration. Peichl et al. [75] compared carbon sequestration within three systems in Ontario, Canada: 13-year-old hybrid poplar shelterbelt plus barley; a 13-year-old Norway spruce (*Picea abies*) shelterbelt plus barley; and a barley-only crop system. Total carbon sequestration was 15.1 and 6.4 Mg C ha$^{-1}$ higher than the barley-only system for hybrid poplar and Norway spruce, respectively. Carbon stock in the soil was also significantly different between the systems: 78, 66, and 65 Mg C ha$^{-1}$ for hybrid poplar, spruce, and barley-only systems, respectively.

Amichev et al. [74] and Amichev et al. [77] used the 3PG and CBM-CFS3 models to quantify tree growth and carbon stocks of shelterbelts. The CBM-CFS3 model was originally developed for the Canadian forest industry sector and has been used at various scales of analysis, from stand to landscape levels, to simulate forest stand growth and carbon dynamics. Similarly, 3PG is a hybrid model, designed to model forest growth, which was also adapted for use in shelterbelt systems [73,76]. Amichev et al. [74] parametrized 3PG to quantify carbon stocks of white spruce shelterbelts in a large scale study extending across five soil zones in Saskatchewan and spanning several decades of planting, from 1925 to 2009. They estimated the total above-ground biomass at 117.6 Mg C km$^{-1}$, ranging from 106 to 195 Mg C km$^{-1}$, depending on the soil zone. The total ecosystem carbon flux increased from 0.33 to 4.4 Mg C km$^{-1}$ year$^{-1}$, from year 1 to 25, reaching a peak of 5.5 C km$^{-1}$ year$^{-1}$ Mg 39 years after planting. Average-stand biomass at age 60 was 241.3, 238.6, and 227.3 Mg km$^{-1}$ at 2.0, 3.5, and 5.0 m tree spacing design, respectively. Overall, total carbon stocks for all white spruce shelterbelts in the province, planted over the span of eight decades, was 50,440 Mg C, sequestered in over 991 km of planted shelterbelts.

Using the same methodology, Amichev et al. [20] estimated the growth of five additional common shelterbelt species planted across Saskatchewan—hybrid poplar, Manitoba maple, Scots pine, green ash, and Caragana planted between 1925 to 2009, in three spacing designs (2.0, 3.5, and 5.0 m), and at four different mortality rates (0, 15, 30, and 50%). They estimated total shelterbelt carbon stocks for the province at 10.8 Tg C. Overall, the average carbon sequestration rate on a length basis (per km) was estimated 1.73 to 6.54 Mg C km$^{-1}$ year$^{-1}$. The carbon sequestration rates for the individual species were 6.03–6.54 Mg C km$^{-1}$ year$^{-1}$ for hybrid poplar, 1.73–2.03 Mg C km$^{-1}$ year$^{-1}$ for caragana, 1.90–2.17 Mg C km$^{-1}$ year$^{-1}$ for Scots pine, 2.43–2.75 Mg C km$^{-1}$ year$^{-1}$ for white spruce, 1.78–1.98 Mg C km$^{-1}$ year$^{-1}$ for green ash, and 2.39–2.60 Mg C km$^{-1}$ year$^{-1}$ for Manitoba maple shelterbelts. The per-unit-area C rates (Mg C ha$^{-1}$ year$^{-1}$) represent the C sequestration rate across 1-ha cumulative land area located directly underneath the shelterbelt tree crowns. These C sequestration rates included the C locked in live and dead above- and below-ground biomass (i.e., stems, branches, leaves, roots),

as well as litter layer (i.e., decomposing tree branches and leaves) on the soil surface, and soil organic matter added into the soil.

The Holos model is an empirical, process-based, farm-scale model that estimates GHGs emissions from farms based on site-specific input information [37,76]. The model relies on details such as enteric fermentation, manure management, cropping systems, energy use, and presence of planted trees [76]. Researchers have used the Holos model for exploration of diverse farming scenarios through many simulations, aiming for minimal GHGs releases from a farm. For example, Holos was used by Amadi et al., [38] to calculate the potential of hybrid poplar, white spruce and caragana shelterbelts to offset GHGs emissions from cereal (*Triticum aestivum* and *Avena sativa*) production for a 60-year simulation period, at five shelterbelt planting densities. At the highest planting density (i.e., 5% of total farm area occupied by trees), hybrid poplar, white spruce, and caragana shelterbelts reduced farm GHGs emissions by 23%, 18% and 8%, respectively. The majority of this GHGs offset (95%) was attributed to C sequestered in wood biomass and the soil. The rest was attributed to lower $N_2O$ emissions and $CH_4$ oxidation, commonly observed within the shelterbelt zone. For a 60-yr simulation, the estimated carbon stocks were 8712, 5581, and 1705 Mg C (at the most-dense spacing) for hybrid poplar, white spruce, and caragana, respectively.

Likewise, statistical models also have been used to estimate carbon sequestration in shelterbelts [10,50]. Possu et al. [78] assessed 15 allometric models on Ponderosa pine windbreaks and used the best model to estimate carbon sequestration for 16 shelterbelt tree species in Nebraska, projected over 50 years in nine areas of the U.S. They found that carbon sequestration potential ranged from $1.07 \pm 0.21$ to $3.84 \pm 0.04$ Mg C ha$^{-1}$ year$^{-1}$ for conifer species and from $0.99 \pm 0.16$ to $13.6 \pm 7.72$ Mg C ha$^{-1}$ year$^{-1}$ for broadleaved deciduous species. Zhou et al. [9] assessed carbon stocks in shelterbelts in Montana, U.S., with two types of statistical models: precise preferred models which required more variables from expensive data inventories; and cost preferred models, which were simpler to fit, but were less precise. The authors concluded that both sets of equations were effective to estimate shelterbelt biomass and that the precision preferred models were between 0.8 and 1.2% more precise than the cost preferred models. They found that above-ground biomass for a single row of Russian-olive shelterbelt was 110 metric tonnes km$^{-1}$ (110 Mg km$^{-1}$), that converted to approximately 55 Mg C km$^{-1}$, 60 years after planting (equal to sequestration rate of 0.91 Mg C km$^{-1}$ year$^{-1}$).

### 3.1.2. Carbon Stocks Below-Ground

Soil is the biggest organic carbon pool on Earth [58]. It is calculated that the world's agricultural and degraded soil are able to sequester 50 to 66% of all carbon released, equivalent to 42–78 gigatons of carbon [79]. The two major below-ground carbon pools are the soil organic carbon (SOC) and below-ground biomass (i.e., fine and coarse roots). However, even though the below-ground carbon sequestration potential is known, the methods to quantify it are still in their infancy and there is no established standard protocol to follow when measuring it worldwide.

For determining SOC, the two main problems are the lack of standards for soil aggregate class definitions and what soil depth to sample [62]. Aggregates are often classified according to their ability to resist slaking in water, and fortunately, a trend in new studies is starting to follow a standard (<53 μm, 53–250 μm, and <250 μm) [62]. Soil depth is the most serious issue on assessing and comparing underground carbon sequestration/stocks [62]. Most studies sample up to 20 or 30 cm depth [62]. For carbon stock studies in agroforestry, assessing soil to a greater depth is extremely important, since the sub-soil is a crucial part of carbon stabilization [62]. A general technique that is simple and practical for any situation is needed in order to allow for comparisons among studies, facilitating better whole system shelterbelt carbon estimation. This is extremely important, given that above-ground biomass alone represents just one pool of the carbon sequestered in shelterbelt agroforestry systems and all ecosystem components need to be considered. Soil carbon sequestration is estimated to be around two-thirds of the whole carbon sequestered in the ecosystem [80,81]. For example, Chu et al. [72], studying shelterbelt trees planted by the Three-North Shelterbelt Program, found that 67% of the

carbon was stored in the soil, with roots representing 13%, and above-ground biomass representing only 10%.

The SOC pool is in constant interaction with other C pools in shelterbelt systems, receiving inputs from above- and below-ground system components. For example, above-ground inputs include litter fall (i.e., fallen leaves and branches), animal excrements, and decomposed biomass. Below-ground carbon inputs include root litter and rhizosphere depositions [35]. Higher SOC inputs help to maintain soil moisture and fertility, and is strongly affected by precipitation, temperature, soil texture, average stem and crown diameter, tree height, amount of surface litter, and shelterbelt species and age [28,35].

Below-ground biomass measurement techniques for shelterbelt systems also have not been thoroughly explored, since they tend to be resource demanding and time consuming, and there is no well-established methodology to sample the below-ground system. Because of this, comparison between studies is problematic [37,62,66,74,80]. Aiming to facilitate below-ground carbon estimation, the IPCC recommended below-ground biomass estimations using established relationships with above-ground biomass [55]. Similarly, Kort and Turnock [73] recommended root biomass estimations to be done by considering constant ratios of 40%, 30% and 50% of above-ground biomass for deciduous, coniferous, and shrub shelterbelts, respectively, which were also prescribed by Freedman et al. [79] and Grier et al. [82]. However, this method is problematic, since root systems vary with species, climatic zone, and environmental conditions within the region [62]. To be more comprehensive, more methods need to be tested for many species across many site conditions. For example, a dry environment would stimulate a deeper root system, while a less dry environment would produce shallower roots for the same tree species.

Numerous studies have observed a trend of soil carbon loss occurring when a new shelterbelt is first planted, most likely due to site preparation and land use change, which is offset years later, as the trees grow more extensive roots systems [28,35,74,80]. This is a natural process that usually takes place when land use is changed. For example, when a natural ecosystem is replaced by agriculture, around 60% and 75% of SOC is lost in temperate and tropical climate, respectively [80]. Amichev et al. [74] found that soil carbon stocks decreased during the first 10 years following shelterbelt implementation, losing about 3.5% within the first five years. Their model simulations illustrated that carbon emissions due to land cover change were completely offset by the ages of 17, 18, and 21 for shelterbelts planted at 2.0, 3.5, and 5.0 m spacing, respectively.

Even though it takes years to compensate carbon loss due to shelterbelt planting, in dry environments, such as the Prairies, where biomass production is not very high, shelterbelts can be an important source of organic matter for the soil. Shelterbelts increase SOC, moisture, and fertility, and consequently, increase carbon sequestration into the soil pool. Research has demonstrated that SOC under shelterbelt trees canopies is greater than SOC under crops, and that this difference tends to be less pronounced in deeper layers of the soil [28,83], but vary according to the species considered [36]. For example, Amadi et al. [28] studying the carbon sequestration potential of different shelterbelt species in Saskatchewan in the 0–7.5 cm and 7.5–15 cm soil layers, reported higher SOC stocks within the top soil layer. Similar were the results reported by Dhillon and Van Rees [35], who studied the SOC sequestration potential of six common shelterbelt species planted in Saskatchewan (green ash, hybrid poplar, Manitoba maple, white spruce, Scots pine and caragana), ranging from five to 63 years of age. Their results showed that soil organic matter concentration was 30% greater under shelterbelts than adjacent cropped fields, and that the SOC stock was 19% greater under shelterbelts than under crops. This difference is due to lower bulk density of soils under shelterbelts than under cropped fields. This lower bulk density is attributed to the presence of organic matter, extensive tree root systems, and due to the absence of heavy machinery traffic for the years since the shelterbelt was implemented. This finding was corroborated by Sauer et al. [80] who also reported lower bulk density under shelterbelts than under the adjacent cropped field. The bulk density in this case was 13% lower in the 0–10 cm layer, and 7% lower in the 10–30 cm layer. They also found that soil under shelterbelts had 18.6 Mg C ha$^{-1}$ more SOC than the soil under crop production within the top 50 cm of soil, and

that litter under shelterbelts contained an additional 3–8 Mg C ha$^{-1}$. The SOC stocks vary greatly by species because of differences in litter composition leading, to differences in litter decomposition rates. For example, these authors found that white spruce shelterbelts had 20.8 g C kg$^{-1}$ more SOC in the 0–5 cm soil layer than the adjacent cropped field, while green ash had only 0.8 g C kg$^{-1}$ more SOC than the adjacent cropped field.

An important aspect of the shelterbelt SOC pool is the long residence time, which emphasizes the shelterbelts' role as an effective climate mitigation tool from a global perspective. Needless to say, the longer the added soil carbon remains in the soil pool, the better. Organic carbon compounds in the soil can be classified as either labile, with residence time in the soil of a few months, or as recalcitrant, with residence time in the soil of a few decades. Dhillon and Van Rees [35] analyzed the effect of shelterbelts and cropped fields in Saskatchewan on the distribution of soil organic carbon density fractions. They found an increase in the SOC labile light fraction (71%) and the stable heavy fraction (22%) for soils under shelterbelts compared to cropped fields. The majority of SOC added in the 0–10 cm layer belonged to the labile light fraction, and the majority of the SOC added in the 10–30 cm layer belonged to the heavy fraction. The SOC light fraction was generally associated with conifer shelterbelts, whereas the SOC heavy fraction was associated with deciduous trees. For example, Manitoba maple litter was abundant with more resistant forms of soil organic matter (i.e., needing more time to decompose) [61].

Shelterbelt management practices also influence SOC stocks, and more specifically, the type of soil organic matter compounds, and consequently their residence time within the soil pool. The chemical composition of these soil compounds affects microorganisms-enzymes interactions and determine their stability and residence time in the soil [83]. The soil under shelterbelts have more processed C forms, such as aliphatic C, aromatic C, and ketones, that are harder for microbes to break down, while the soil under cropped fields have more sugars and alcohols [62,63].

Soil greenhouse gases flux studies are also important to better understand below-ground carbon dynamics. Amadi et al., [37] compared soil $CO_2$, $CH_4$ and $N_2O$ fluxes in shelterbelt systems with adjacent cropped field in Prince Albert, Saskatchewan, using non-steady state vented chambers. Even though they found greater $CO_2$ fluxes under shelterbelts than in crop fields (probably due to higher microbial activity, root respiration and litter decomposition), soil organic carbon under the shelterbelts was 27% greater than under the adjacent cropped field. Shelterbelts contributed to the offset of other GHGs released from farming activities, not just $CO_2$, by enlarging the $CH_4$ soil sink, absorbing 58% and 81% more $CH_4$ than soil in cropped fields, in 2013 and 2014, respectively. Cumulative seasonal $N_2O$ emissions from shelterbelt areas were two- to five-times lower than emissions by cropped fields nearby. A similar study by Amadi et al. [28] investigated whether a two-row 31-year-old hybrid poplar-caragana shelterbelt influenced the soil organic carbon and GHG flux dynamic on its surrounding area, compared to an adjacent cropped field. They found that soil organic carbon concentration in the soil was greater in the proximity to shelterbelts, and that $CH_4$ uptake decreased with increasing distance away from the shelterbelt. The $N_2O$ release was smaller under the shelterbelt and increased towards the cropped field, and the $CO_2$ flux between soil and atmosphere was more intense in the proximity to shelterbelts, which was in agreement with Amadi et al. [37]. Higher fluxes were attributed to higher organic matter concentration, microbial activity, and tree root respiration in the proximity of the shelterbelts [28,37].

## 4. Conclusions

This review paper summarized the currently available research-based knowledge surrounding shelterbelt agroforestry systems, and aimed to increase the awareness of researchers, farmers, industry, and policymakers of the climate change mitigation potential of planted shelterbelts throughout Canada and the world. The current knowledge-base clearly indicates that shelterbelts have a great potential for carbon sequestration, both in above- and below-ground pools of the system. As the trees in the shelterbelts continue to grow, they are able to reduce and offset a significant portion of the carbon

released from agricultural practices, while providing several other social and environmental benefits, both for the public and private sectors.

In order to preserve existing shelterbelts, and promote the planting of new ones, new effective policies are needed in Canada that would provide farmers with the necessary economic incentives, and cost recovery for shelterbelt establishment and maintenance. This is especially important in the post-Paris Accord era, as the Government of Canada is steadily transitioning towards a carbonless economy, at the forefront of which are the farming communities in the Canadian Prairies. To better understand the budgetary impact of a carbonless economy on the Canadian farmer, whole-farm cost analysis studies, and shelterbelt decision-support tools for farmers that account for shelterbelt establishment and decades-long maintenance costs (i.e., time, machinery, and labor), will be needed. New policies that will help farmers meet shelterbelt-related costs will likely have a significant impact on the carbon mitigation potential of these systems in the long term. All these actions, if executed and coordinated effectively, can provide a major step for Canada, and for the world, towards a truly decarbonized economy.

**Author Contributions:** Conceptualization, R.C.M., C.P.L., B.Y.A., K.V.R.; writing—original draft preparation, R.C.M.; writing—review and editing, B.Y.A., K.V.R., R.C.M., C.P.L.; supervision, C.P.L.

**Funding:** The authors gratefully acknowledge the financial support given by the Agricultural Greenhouse Gases Program, Government of Canada (Grant #AGGP2-017).

**Conflicts of Interest:** The authors declare no conflicts of interest.

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
