# Peer review of "Above- and Below-Ground Carbon Sequestration in Shelterbelt Trees in Canada: A Review"

_forests, doi:10.3390/f10100922_

Round 1

Reviewer 1 Report

1. Overview: shelterbelts around the world

The review is complete and it cites the most relevant information in this field

2. Shelterbelt history in the Canadian Prairies

The authors made a comprehensive revision about the background of the shelterbelts

3.Environmental services provided by shelterbelts

The authors put relevant information about shelterbelts environment services,

4. Mapping and Inventory

This is a good topic and it is sound and well analyzed.

5. Carbon sequestration

This topic can be joined to the topic three “Environmental services provided by shelterbelts”

Minor Comments

some problems with the carbon units, specifically with power or exponent (i.e Mg C km -1). Make sure the manuscript keeps the original style.

Reviewer 2 Report

Overall comments:

The paper aims to review shelterbelt C  sequestration in Canada. The history of shelterbelts in Canada, included in the review, is its strongest and most unique part. Overall however, the review is not very in-depth and its structure is not very solid, jumping from overall level to Canada and back, and from methods to results and back. I suggest overall re-write.

Currently, no aspect of the shelterbelts aside from their history in Canada is well reflected upon, nor is the situation/results in Canada looked at in the larger context or in comparison to elsewhere in the world. No information is drawn or compared to related fields in agroforestry. C sequestration potential of shelterbelts is not discussed in comparison to other C sequestration measures in agriculture or forestry. The lifespan of shelterbelt, the extent of C sequestration into the future, re-planting or regeneration are not discussed. Planting a shelterbelt could be more analytically looked upon in the context of land-use change, which starts from and leads to semi-steady system, each with its own characteristic level of semi-steady C stock. C sequestration effects come largely from this land-use change, unless potential sustainable use of timber or biomass is taken into account. Thus C seq does not necessarily continue until perpetuity and life-span should be discussed.

Definitions of key terms, such as shelterbelt, are missing. What is a shelterbelt, how tall, wide, how maintained, how long lifespan, is alley cropping a shelterbelt, is pruned hedge a shelterbelt, is windbread a shelterbelt, why/why not. Shelterbelt effects are discussed predominantly on km-1 of row basis. It is not clear however, how much shelterbelt is feasible per km2 of fields. Potential negative effects on crop yield through shading and competition for water are not discussed, nor the optimal spacing, height, species or management to keep the effects of shelterbelt positive for crop yield. There is a clear need to solidify the structure of the review. Methods used in shelterbelt C sequestration research could either be left out or separated to a chapter/chapters, instead of being scattered around the review.

Comments by line/chapter

Lines 10-29, Summary. Does not reflect the contents of the review. Focuses on history, which makes up only one page of the review.

44-46. How much is the increase in proportion to forestry or agriculture publications overall? Number of publications in most fields has increased due to increased no. of journals and pressure to publish. Increase needs to be shown in comparison to some relevant field, not just absolute no. of papers. Is it enough to search for "shelterbelt"? is the term universally used and contains all relevant studies?

56, 142, and elsewhere. Reference no.10 is very prominent and oftentimes the only reference, especially for some piece of numeric information. Yet, based on the list of references, it is not a research paper. Where are the numbers actually coming from?

74-76. The logic between farmers underestimating benefits and possible government rewards is not solid. Rather, the farmers do not benefit from C seq. currently, independent on how they value it.

83-134. Interesting historical information that I had not come across previously, written out quite nicely. The highlight of the review.

98: Acre in SI units?

168-177: shelterbelt effects on crop water use. Please find more studies. Kort’s paper also identifies pitfalls and cases when shelterbelts do not help the yield. Please discuss negatives and how to avoid these. If you do not want to discuss spacing, orientation etc., then make clear these are important, and refer the reader to go elsewhere for more info.

190-192: internal comment, should be deleted?

194 and onwards, mapping and inventory. This is the only chapter focusing on methods, apparently only in a handful of Canadian studies. Why are these included and others not? What is the point of reviewing these methods in some detail, when biomass and soil C methods are included only in passing? Very unbalanced (see overall comments).

321, chapter on belowground C stocks. Chapter numbering is off (5.2 --> 5.3). Soil depth studied is often rather shallow, which leads to issues with stock estimates. This uncertainty should be discussed. Chapter focuses on C stocks and takes little interest in C sequestration, which is the headline of the paper. There are much less studies on C sequestration underground, but based on the data collected, some discussion on the topic can and should be included. It could include e.g. discussion on why there are little studies and what could be done to rectify this.

330-335. How reliable can it be to estimate root biomass from root-shoot ratio, with varying species and growing conditions? please discuss.

404-422 largely lists results from a few papers, and does not discuss the reasons or mechanisms, unlike some of the reviewed studies. Results are not compared to any context in e.g. elsewhere in the world or other agroforesty systems. Please be more analytical – that is where the value of a review paper lies.

Round 2

Reviewer 2 Report

Revisions requested were major, and it is clear there has not been time to make the paper much more in-depth or inclusive, so it's value has not increased very much. The language of the initial submitted paper was good, but the revised sections in this version are very poor English. Language review in these sections, which are relatively short, is needed.

Previous comment on defining shelterbelt: the relation of shelterbelts to hedges (used predominantly in UK, France, Central Europe) still missing. Are they excluded here, and if, on what basis. If not excluded on reasonable basis, must be included.

Table 2. Is the C sequestration potential of shelterbelts given per hectare of field which has shelterbelt as a part of it, or just the shelterbelt itself (as seems to be the case for Amichev et al. data)? So is the cropland left out?
If it is per area of shelterbelt only, then the comparison to other systems is not appropriate without considering how much crop/shelterbelt there is per hectare, as many of the other systems C seq values appear to include both woody vegetation and the crop. All values in the same column should be comparable, if not, in separate columns with informative headings or otherwise separated.

L. 852-862 almost incomprehensible language.

L. 863-867 Why mention this method, when the paper is not about methods? remove or attach to proper context.

L. 710-714. I assume this was added upon the comment about different measuring depths in the literature (point 10). It is too general however, not even mentioning sampling depth or giving any consideration to what kind of error/bias cold result from comparing disparate soil depths.

Point 10, second paragraph in the initial comments, on the chapter on belowground C. How about making the chapter name more descriptive to its contents, if there is little info on sequestration.
